# Evidence of Transmission of *Plasmodium vivax* 210 and *Plasmodium vivax* 247 by *Anopheles gambiae* and *An. coluzzii*, Major Malaria Vectors in Benin/West Africa

**DOI:** 10.3390/insects14030231

**Published:** 2023-02-25

**Authors:** Razaki A. Ossè, Filémon Tokponnon, Germain Gil Padonou, Mariette E. Glitho, Aboubakar Sidick, Arsène Fassinou, Come Z. Koukpo, Bruno Akinro, Arthur Sovi, Martin Akogbéto

**Affiliations:** 1Ecole de Gestion et d’Exploitation des Systèmes d’Elevage, Université Nationale d’Agriculture, Kétou BP 44, Benin; 2Centre de Recherche Entomologique de Cotonou, Ministère de la Santé, Cotonou 06 BP 2604, Benin; 3Ecole Polytechnique d’Abomey Calavi, Université d’Abomey-Calavi, Abomey-Calavi 01 BP 526, Benin; 4Faculté des Sciences et Techniques, Université d’Abomey-Calavi, Abomey-Calavi 01 BP 526, Benin; 5Faculté d’Agronomie, Université de Parakou, Parakou BP 123, Benin; 6Faculty of Infectious and Tropical Diseases, The London School of Hygiene and Tropical Medicine, London WC1E 7HT, UK

**Keywords:** *P. falciparum*, *P. vivax* 210, *P. vivax* 247, *Anopheles gambiae*, *An. coluzzii*, malaria, Benin

## Abstract

**Simple Summary:**

For any country to achieve an effective malaria control program, the information on causative parasite diversity and dynamics in the region is a key factor. This study demonstrates that in Benin, the occurrence of malaria cases is not solely caused by *Plasmodium falciparum* (common malaria parasite) and involves other non-falciparum species. The aim of this work is to assess the prevalence of various *Plasmodium* species in locally collected mosquito cohorts, either through human landing catches or pyrethrum spray catches. Thus, the comparative data on the prevalence of circumsporozoite protein (CSP) antibodies of *P. falciparum* and *P. vivax* in *Anopheles gambiae* s.l., the major insect-vector responsible for transmitting the malaria parasite to humans in Benin, were reported. Further, to delineate the possible contribution of various *Plasmodium* species in malaria infection in collected mosquitoes, a molecular species-specific polymerase chain reaction analysis was performed. The important finding of this study reveals the variation in the distribution of *Plasmodium* species prevalence in Benin. The information incurred through this study will be very helpful for malaria control stakeholders as well as planning and management agencies designing effective diagnostics and preventive measures and treatments to curb the lethal impact of malaria in Benin.

**Abstract:**

Current diagnostic and surveillance systems in Benin are not designed to accurately identify or report non-*Plasmodium falciparum* (*Pf*) human malaria infections. This study aims to assess and compare the prevalence of circumsporozoite protein (CSP) antibodies of *Pf* and *P. vivax* (*Pv*) in *Anopheles gambiae* s.l. in Benin. For that, mosquito collections were performed through human landing catches (HLC) and pyrethrum spray catches (PSC). The collected mosquitoes were morphologically identified, and *Pf*, *Pv* 210, and *Pv* 247 CSP antibodies were sought in *An. gambiae* s.l. through the ELISA and polymerase chain reaction (PCR) techniques. Of the 32,773 collected mosquitoes, 20.9% were *An. gambiae* s.l., 3.9% *An. funestus* gr., and 0.6% *An. nili* gr. In *An. gambiae* s.l., the sporozoite rate was 2.6% (95% CI: 2.1–3.1) for *Pf*, against 0.30% (95% CI: 0.1–0.5) and 0.2% (95% CI: 0.1–0.4), respectively, for *Pv* 210 and *Pv* 247. *P. falciparum* sporozoite positive mosquitoes were mostly *An. gambiae* (64.35%), followed by *An. coluzzii* (34.78%) and *An. arabiensis* (0.86%). At the opposite, for the *Pv* 210 sporozoite-positive mosquitoes, *An. coluzzii* and *An. gambiae* accounted for 76.92% and 23.08%, respectively. Overall, the present study shows that *P. falciparum* is not the only *Plasmodium* species involved in malaria cases in Benin.

## 1. Introduction

Globally, the number of malaria cases was estimated at 241 million in 2020 versus 227 million in 2019 [1]. WHO African Region shoulders the heaviest burden of the disease, with 95% of all cases recorded in 2020 and 602,000 deaths, 80% of which occurred in children under 5 years old [1]. The intertropical zone to which Benin belongs to is a region with an environmental and health situation that is essentially characterized by various tropical diseases, of which the most important is malaria [2]. According to the National Health Information and Management System of the Ministry of Health, malaria constitutes the main reason for consultation and hospitalization in the general population, particularly in children under five. The incidence of the disease in 2020 was 19% in the general population and 41.6% in children under five [3].

The parasite responsible for malaria infection is a protozoan belonging to the genus *Plasmodium.* It is mainly transmitted by *Anopheles gambiae* s.l. (sensu lato), *Anopheles funestus,* and *Anopheles nili* groups [4,5]. Of all the known *Plasmodium* species, 11 (*Plasmodium falciparum*, *Plasmodium vivax*, *Plasmodium ovale*, *Plasmodium malariae, Plasmodium knowlesi, P. cynomolgi, P. inui, P. coatneyi, P. inui*–like, *P. simiovale*, and *P. simium*) are able to infect humans [6,7,8]. The two most widespread species, namely *P. falciparum* and *P. vivax* [1], can live in sympatry [9]. However, of the two, *P. falciparum* is the most virulent one, as it causes 99.7% of malaria cases in the WHO African region [1]. *P. vivax* is widespread in Southeast and Central Asia, as well as South America, but it is present only in a few African countries, including Uganda and Ethiopia [1,10,11,12]. Two other species, including *P. ovale,* which is found mainly in West Africa, and *P. malariae,* which is found in different places globally, are generally very rare [13]. For *P. knowlesi*, the transmission is zoonotic, and parasitism is asymptomatic with low levels of intensity, causing mild disease in humans [14]. Others zoonotic malaria caused by *P. cynomolgi*, *P. inui*, *P. inui*–like, *P. coatneyi*, and *P. simiovale* were detected in Malaysia [7].

Malaria in sub-Saharan Africa has historically been almost exclusively attributed to *P. falciparum*. A key difference between *P. falciparum* and *P. vivax* is the ability of the *P. vivax* to form dormant liver stages (hypnozoites) that evade the human immune system so that it can trigger relapses of clinical episodes several weeks or months after treatment [15]. In Benin, as in many other African countries, several trials focused only on *P. falciparum* in mosquito vectors. This allowed a better knowledge of this parasite at the expense of *P. vivax*, which has particular biological characteristics that allow it to resist prevention methods [16]. However, *P. vivax* infections have been documented in Africa in Duffy-negative blood group individuals who were considered as naturally protected against *P. vivax* infection [17]. Indeed, the Duffy antigen has been identified as an obligatory trans-membrane receptor for *P. vivax* in red blood cells [18]. This implies that the prevalence of *P. vivax* in Africa is probably underestimated, requiring large-scale epidemiological studies to appreciate the burden of *P. vivax* infections [19].

The results from the study conducted in Benin show a surprisingly high exposure of populations to *P. vivax*, resulting in submicroscopic infections [16]. This suggests a likely underestimated and insidious parasite presence in West Africa. While vaccination campaigns and therapeutic efforts are all focused on *P. falciparum*, it is also essential to consider the epidemiological impact of *P. vivax* and test mosquitoes for the presence of antibodies of this *Plasmodium* species. The present study aims to evaluate the prevalence of sporozoite of *P. falciparum* and *P. vivax* in *Anopheles gambiae* s.l., the main malaria vector in different communes throughout Benin.

## 2. Materials and Methods

### 2.1. Study Sites

Mosquito collection was conducted in all 12 regions of Benin from April to December 2019. This collection period coincides with the two rainy seasons of the year, periods characterized by the creation of numerous puddles that favor the proliferation of mosquitoes, including malaria-transmitting ones. In each region, two communes were randomly selected to look for the presence of *Plasmodium* species in mosquitoes. Thus, 24 communes were selected for the study (Figure 1). In each of them, two sites selected at random were surveyed for the collection of mosquitoes.

In communes located in the southern and central part of the country, the hot and humid climate is of the sub-equatorial type with four seasons, including a long (April to July) and short (October to November) rainy season, as well as a long (December to March) and short (August to September) dry season. In the northern communes, the climate is Sudano-Guinean, with a single rainy season (June to November). The rest of the year (December to May) is the dry season. The average annual rainfall is about 1340 mm in the south and 700 mm in the north [20], while the temperature varies little with an annual average between 26 °C and 30 °C [21].

### 2.2. Mosquito Collection

In all sites, the collection of adult mosquitoes was performed using both Human Landing Catches (HLCs) and morning Pyrethrum Spray Catches (PSCs). The two methods were used to increase the likelihood to collect a great number of *Plasmodium* sporozoite-positive mosquitoes.

HLCs were performed between 9:00 p.m. and 5:00 a.m. using eight people, with four (two positioned indoors and two outdoors) used between 9:00 p.m. and 1:00 a.m. and replaced with four others from 1:00 a.m. to 5:00 a.m. Thus, each seated collector used a flashlight as well as a sucking tube to collect the mosquitoes that landed on his/her naked lower legs. Collectors were rotated over collection of hours to limit bias related to individual variations in the attractiveness of *Anopheles* vectors. They were previously trained to collect the mosquitoes.

PSCs were carried out in 10 bedrooms randomly selected in each commune. The houses that were selected have ceilings and do not have eaves. They also have windows and doors that are kept closed before spraying, which prevents mosquitoes from escaping. This method, which was started at 7:00 a.m., enabled the collection of mosquitoes that entered the bedrooms overnight. Thus, the Rambo aerosol (Gongoni Company Limited, Kano, Nigeria) combining Beta-cypermethrin, tetramethrin, and permethrin was sprayed in the interior of rooms where white sheets were previously laid on the floor. This allowed the collection of indoor-resting mosquitoes.

Adult mosquitoes were then morphologically identified using a binocular loupe, as well as the morphological identification key of Gillies & Coetzee [22]. *Anopheles* were stored individually in eppendorf tubes containing silica gel and cotton and kept at −20 °C until processing.

### 2.3. Blood Meal Origin in An. gambiae s.l. Collected by PSC

The origin of blood meal in females *An. gambiae* s.l. collected through PSC in the different communes were determined using the ELISA test, the protocol of which was described by Beier et al. [23]. Antibodies used for this testing were from human (Sigma, St. Louis, MO, USA, 11 mg/mL), cow (Sigma, 11 mg/mL), sheep (Sigma, 4 mg/mL), and pig (Sigma, 11 mg/mL). The blood meal origin was in *Anopheles* to ensure that the detected *Plasmodium* species were from humans living in the commune. Abdomens of blood-fed mosquitoes were ground in 250 μL of Phosphate Buffer Saline (PBS). For each well, 50 μL of each blood eluate were introduced (one mosquito per column) and incubated for 3 h at room temperature. The sensitized plates were washed twice with PBS-Tween 20. For each well and each row, 50 μL of labeled IgG solution were distributed. After a 1 h incubation at room temperature, the plates were emptied and washed with PBS/Tween 20. Then, 100 μL of the peroxidase enzyme substrate were added to each well and incubated in the dark for 30 min. A blue color was obtained, and the reaction was blocked with 50 μL of a 4 N sulfuric acid solution. Results were read at 450–620 nm with the BioTek Elx808 ELISA plate reader (BioTek, Winsooki, VT, USA ). The negative (PBS 1×) and positive controls, in particular the sera of the different hosts sought (human, cow, goat, pig), were placed on each line.

### 2.4. Circumsporozoite Protein ELISA for the Detection of P. falciparum, P. vivax 210 and P. vivax 247

In this study, the detection of *P. falciparum* and *P. vivax* 210 and 247 was done using the heads and thorax of these sampled mosquitoes according to the method described by Burkot et al. [24]. A sampling of the collected *Anopheles* was done by commune in the laboratory before the tests were performed. The heads–thoraxes of the sampled mosquitoes were individually crushed, after a 1 h incubation in 50 μL of blocking buffer (BB) containing IGEPAL CA-630 (Sigma, Darmstadt, Germany). During the grinding, two times 100 μL of BB was supplemented to each ground material. Then, the 96 wells of a plate were first sensitized with circumsporozoite (CS) protein capture monoclonal antibodies (0.5 mg/mL, BEI Resources). For each well of the plate, 50 μL of this solution were introduced and left overnight on the bench. The following day, the wells were emptied without being washed and received 200 μL of BB for 1 h at room temperature. These wells were again emptied without being washed, and each received 50 μL of mosquito ground material. After a 2 h incubation at room temperature, the wells were emptied and washed twice with PBS/Tween 20. Each well then received 50 µL of peroxidase-conjugated monoclonal antibodies (0.5 mg/mL, BEI Resources) for 1 h on the bench. Then, 7.5 µL of the substrate solution was mixed with 5 mL of BB. After 1 h of incubation at room temperature, the wells were emptied and washed 4 times with PBS/Tween 20. Then, 100 μL of the peroxidase substrate were placed in each well. The reaction occurs in the dark for 30 min. At the end, the positive wells were colored in blue. This reaction was blocked with 4 N sulfuric acid. The positive wells were colored in yellow. Plates were read using the BioTek Elx808 ELISA plate reader at 405 nm and 650 nm (BioTek, Winsooki, VT, USA). Laboratory-susceptible strain served as a negative control. *P. falciparum, P. vivax* 210, and *P. vivax* 247 at 2 pg/µL (BEI Resources) were used as positive controls. The testing was simultaneously conducted on 3 plates to detect the presence of the three *Plasmodium* species (*P. falciparum*, *P. vivax* 210, and *P. vivax* 247).

### 2.5. DNA Extraction of Plasmodium Species and Detection of P. vivax Serotypes

To characterize the *Plasmodium* species (*P. falciparum, P. vivax* 210, and 247) in the different communes, *Plasmodium* DNA was extracted from the grindings of the heads–thoraxes of the mosquitoes used for the ELISA CSP testing. In each commune, a total of 50 mosquitoes, including all *Plasmodium* sporozoite-positive samples identified through ELISA CSP testing, plus a subset of randomly selected negative ones, were analyzed by PCR according to the protocol described in Methods in Anopheles Research [25].

*Plasmodium* DNA was extracted with 2% CTAB (Cetyl trimethyl ammonium bromide). The grindings of the heads–thoraxes of the mosquitoes used for the ELISA CSP testing were ground in 200 μL of 2% CTAB. After 5 min in a water bath at 65 °C, the grindings were mixed with 200 μL of chloroform and then centrifuged at 14,000 rpm for 5 min. The supernatant was gently transferred into another tube containing 200 μL of isopropanol then centrifuged at 12,000 rpm for 15 min. The DNA was purified with 200 μL of 70% ethanol. Another centrifugation occurred at 14,000 rpm for 5 min. The content of each tube was gently inverted to preserve the DNA, which will then be dried for at least 3 h on the bench. Finally, 20 μL of bi-distilled water was added to the DNA, which is left on the bench overnight or half a day.

Then, PCR amplification of the *Plasmodium* species was performed in a thermocycler (Eppendorf, Hamburg, Germany) following PCR conditions described in Methods in *Anopheles* Research [25].

The identification of *Plasmodium* species was performed using the primers presented in Table 1. UNR and PLF are universal primers that hybridize to a particular nucleotide sequence in all *Plasmodium* species. FAR forms with UNR and PLF, the pair of primers that amplify a specific nucleotide fragment in *P. falciparum*. The same applies for the pair of primers formed by UNR- PLF and VIR, which amplifies a specific nucleotide fragment in *P. vivax*. Then, the PVCSP1 and PVCSP2 primers were used to identify the *P. vivax* variants (Table 1). DNA of *P. falciparum*, *P. vivax, P. vivax* 210, and *P. vivax* 247 (BEI Resources) was used as positive controls, while sterilized water was used as a negative control. Total of 18 µL of each amplified product for each reaction was analyzed with electrophoresis gel using a 2% agarose gel (Sigma). When successfully amplified, the gel images were recorded and visualized on transilluminator EBOX 1000 (Vilber, Marne-la-Vallée, France).

### 2.6. Molecular Species Identification in the Anopheles gambiae Complex

The abdomens, legs, and wings of the same mosquito individuals previously tested through PCR for detection of *Plasmodium* infection were molecularly speciated using the protocol of Santolamazza et al. [26]. PCR amplification of *Anopheles gambiae* complex species was carried out in a 25 μL reaction mixture containing 5 μL genomic DNA for *An. gambiae* s.l., 1 × PCR buffer 5×, 1.5 mM MgCl_2_, 0.2 mM of each dNTP, 10 pmoles of each primer (Table 1), and 5 U of Taq polymerase (Eurogentec, Liège, Belgium). The amplification was performed in a thermocycler (Eppendorf, Hamburg, Germany) as follows: an initial denaturation of the DNA material at 94 °C for 5 min, followed by 35 cycles of 94 °C for 30 s, 54 °C for 30 s, and 72 °C for 30 s, with a final extension at 72 °C for 10 min. DNA of *An. gambiae*, *An. coluzzii,* and *An. arabiensis* were used as positive controls, while sterilized water was used as a negative control. The amplification product was migrated on a 1.5% agarose gel (Sigma) in the presence of ethidium bromide (Sigma) used as an intercalant. This allows for the assessing of the involvement of each species of the *An. gambiae* complex in the transmission of the different *Plasmodium* species.

### 2.7. Data Analysis

Data were analyzed using R Core Team software (version 4.1.3-2022) and Graph Pad Prism software version 5.0. The graphs were made using Graph Pad software version 5.0. The exact binomial method was used to calculate the prevalence and their confidence intervals with the R Core Team software (version 4.1.3-2022).

### 2.8. Ethical Consideration

The present study received ethical approval from the Institutional Ethics Committee of Centre de Recherche Entomologique de Cotonou (N° 04/IECC). Mosquito collectors gave their consent prior to being involved in the study. They were trained to collect the mosquito before being bitten and were regularly subjected to medical check-ups. When they had malaria, they were taken care of and received anti-malarial medication like Artemisinin-based Combination Therapy if required. They were also vaccinated against yellow fever.

## 3. Results

### 3.1. Mosquito Species Composition

Table 2 shows the species composition of mosquito populations collected through HLCs in the different surveyed regions over the study period. Overall, a great diversity was observed in mosquito fauna collected on human bait, regardless of the commune. In total, 32,773 specimens of mosquitoes belonging to 18 species were collected using HLCs. Overall, the most predominant species in the whole study area were *Culex quinquefasciatus* (40.48%), followed by *Anopheles gambiae* s.l. (20.92%)*, Culex gr decens* (10.23%)*, Mansonia africana* (18.27%), *Anopheles funestus* (3.94%), and *Aedes aegypti* (2.61%) (Table 1). At the region level, *Anopheles gambiae* s.l. was the most abundant mosquito species in Zou (63.9%), Donga (56.9%), Atacora (50.8%), Mono (39.1%), and Couffo (35%), while *Anopheles funestus* was the most abundant one in Collines (60.9%). In the Ouémé, Littoral, Atlantique, and Borgou regions, *Cx quinquefasciatus* was the major species with frequencies of 51.7%, 78.6%, 43.8%, and 48.3%, respectively. *Mansonia africana* was predominant in the Plateau (59%) and Alibori (57.4%) regions (Table 2).

### 3.2. Molecular Species Identification of the Sigbling Species within the Anopheles gambiae Complex

Of the 6857 collected, 940 (13.7%) specimens of *An. gambiae* s.l. were analyzed. *An. gambiae, An. coluzzii,* and *An. arabiensis* were the species recorded and accounted for 53.68%, 45.88%, and 0.44%, respectively (Figure 2). In the Porto-Novo, Aguégués, 3rd district of Cotonou, 9th district of Cotonou, Ouidah, So-Ava, Athiémé, Grand Popo, Aplahoué, and Dogbo regions, *An. coluzzii* was the predominant species with mean frequencies varying between 71.43% and 100%. Conversely, in the regions of Ifangni, Pobè, Agbangnizoun, Ouèssè, Savè, N’dali, Sinendé, Ouaké, Bassila, Kérou, and Kouandé, *An. gambiae* was the major species with a mean frequency ranging from 58.62% to 100%. In the commune of Djidja, the two species were collected in similar proportions. *An. arabiensis* was only found in the commune of Savè (Figure 2).

### 3.3. Blood Meal Origin in An. gambiae s.l. Collected through PSC

Of the 1392 mosquitoes collected by PSC, 514 were specimens of *An. gambiae* s.l., of which 311 (60.5%) were blood-fed. Out of the 311 blood-fed specimens of *An. gambiae* s.l. that were tested through ELISA blood meal in the study area, 95.5% took their blood meal on humans (Table 3).

Approximately 16.67% and 7.14% of the tested individuals of *An. gambiae* s.l. had successful blood meal intake on cow in the communes of Kouandé and Karimama, respectively. In Athiémé and Karimama, the proportion of of *An. gambiae* s.l that blood-fed on sheep was 4.87% and 3.57%, respectively. Blood meal intake on pigs occurred in 1.69%, 8.33%, and 1.61% of specimens of *An. gambiae* s.l analyzed in Aguégués, Sô-Ava, and Athiémé, respectively. In the commune of Sô-Ava, 4.17% of tested individuals of *An. gambiae s.l* took their blood meal on both humans and pigs (Table 3).

### 3.4. Prevalence of P. falciparum, P. vivax 210 and 247 CSP Antibodies in An. gambiae s.l.

#### 3.4.1. Plasmodium Detection by ELISA

A total of 3300 head–thoraxes of *An. gambiae* s.l. out of the 6857 collected were tested through ELISA CSP for each of the three *Plasmodium* species, equating to a total of 9900 samples analyzed. The number of sporozoite-positive mosquitoes for each *Plasmodium* species at the commune level is mentioned on Figure 3. The combined data showed that 86 mosquitoes were positive for *P. falciparum*, 9 for *P. vivax* 210, and 7 for *P. vivax* 247. Thus, the average prevalence (sporozoite index: SI) for *P. falciparum* was 2.6% (CI: 2.1–3.1) compared to 0.3% (CI: 0.1–0.5) for *P. vivax* 210 and 0.2% (CI: 0.1–0.4) for *P. vivax* 247 in the whole study area.

Out of the 228 specimens of *An. funestus* analyzed, only two (one in Ouèssè and one in Savè) were positive for *P. falciparum*. No specimen of *An. funestus* was found positive for *P. vivax* 210 and *P. vivax* 247.

The three species of *Plasmodium*-*P. falciparum*, *P. vivax* 210, and *P. vivax* 247-were found in *An. gambiae* s.l in Cotonou with a respective prevalence of 0.7% (CI: 0–2.1), 0.55% (CI: 0.05–3.5), and 1.45% (CI: 0.15–5). In the communes of Aguégués, Agbangnizoun, and Sinendé, *P. falciparum* (SI = 2% CI: 0.3–3.7; 2.8% CI: 0–6.6; and 1.5% CI: 0–4.3, respectively) and *P. vivax* 210 (SI = 1.2% CI: 0.2–3.5; 1.4% CI: 0–7.5 and 1.5% CI: 0–7.9, respectively) were detected in *An. gambiae* s.l. during the study. In Sô-Ava and Dogbo, *P. falciparum* (SI = 0.5% CI: 0–1.5 and 6% CI: 1.3–10.7, respectively) and *P. vivax 247* (SI = 0.5% CI: 0–2.8 and 2% CI: 0.2–7, respectively) were detected (Figure 3). In Ouidah, only *P. vivax* 210 (SI = 1.2% CI: 0.2–4.4) was found in *An. gambiae* s.l., whereas *P. falciparum* was the only species detected in Porto-Novo (1.5% CI: 0–3.2), Ifangni (5% CI: 2–8.1), Djidja (8% CI: 3.7–12.3), Savè (0.9% CI: 0–2.1), Ouèssè (4% CI: 1.3–6.7), Athiémé (0.3% CI: 0–1), Kérou (4.7% CI: 1.3–8), Kouandé (6.7% CI: 2.7–10.7), Ouaké (4.6% CI: 1–8.2), Bassila (5% CI: 1.4–8.6), Karimama (2.7% CI: 0–6.3), and N’dali (1.5% CI: 0–4.5).

No mosquito was found positive for the three searched *Plasmodium* species in Pobè, Grand Popo, Aplahoué, and Banikoara. No co-infection was recorded in the *Anopheles* tested throughout the surveyed communes (Figure 3).

#### 3.4.2. Plasmodium Detection by PCR

All samples positive for any of the *Plasmodium* species sought by the CSP ELISA test were also positive for the PCR test (Figure 3). Of the 940 samples of *An. gambiae* s.l. tested through PCR, 115 were positive for *P. falciparum*, 13 for *P. vivax* 210, and 9 for *P. vivax* 247 (Figure 3, Figure 4, Figure 5). *P. falciparum* was detected in all surveyed communes except Pobè, where no positive specimen was found for the three searched *Plasmodium* species.

*P. vivax* 210 was detected in Cotonou (2/100), Ouidah (2/50), Porto-Novo (1/50), Aguégués (4/50), Athiémé (1/50), Agbangnizoun (1/33), Karimama (1/32), and Sinendé (1/32), whereas *P. vivax* 247 was found only in Cotonou (4/100), Sô-Ava (3/50), and Dogbo (2/38). In Ifangni (13/50) and Grand Popo (2/50), *P. falciparum* was the sole detected *Plasmodium* species.

As observed with the CSP ELISA test, no co-infection was recorded. In Grand Popo, Aplahoué, and Banikoara, *P. falciparum* sporozoite-positive mosquitoes were found through PCR, which was not the case with the ELISA CSP test. The same trend was observed with *P. vivax* 210 in Karimama.

### 3.5. Distribution of the Plasmodium Species in the Sibling Species of the Anopheles gambiae Complex

Among the 115 specimens of *Anopheles gambiae* s.l found positive for *P. falciparum* through PCR, 64.35% (*n* = 74) were *An. gambiae,* 34.78% (*n* = 40) were *An. coluzzii,* and 0.86% (*n* = 1) were *An. arabiensis*. Similarly, the *P. falciparum* infectivity rate was significantly higher in *An. gambiae* than in *An. coluzzii* (*p* < 0.001).

At the opposite, most *P. vivax* 210-positive mosquitoes were *An. coluzzii* (76.92%). The rest were *An. gambiae*. The infectivity rates of *P. vivax* 210 were significantly higher in *An. coluzzii* than in *An. gambiae* (*p* < 0.05).

Concerning *P. vivax 247*, only *An. coluzzii* was positive to this variant (Table 4).

## 4. Discussion

In the current context of malaria elimination, the identification of *Plasmodium* species in humans and vectors is a prerequisite not only to understand the epidemiology of the disease, but also to implement an effective control strategy against these parasites. The present study shows the presence of antibodies of *P. falciparum* and *P. vivax* in the *An. gambiae* complex, the main malaria vector complex in Benin. This result corroborates the work of Poirier et al. [16], who showed a surprisingly high exposure of Beninese to *P. vivax*, resulting in submicroscopic infections.

In the present study, 18 mosquito species, including five *Anopheles* ones, were identified in the 24 surveyed communes. These results are reminiscent to those of Huttel [27] and Salako et al. [28], who recorded 13 and 14 mosquito species in Southeastern and Northwestern Benin, respectively. Of the five collected *Anopheles* species, three (*An. gambiae* s.l., *An. funestus gr.,* and *An. nili gr*) have already been incriminated in malaria transmission in Benin [4,5,29,30,31,32].

The molecular species identification performed in *An. gambiae* s.l. revealed the presence of three sibling species, namely *An. gambiae*, *An. coluzzii,* and *An. arabiensis*. These species were found in sympatry but in variable proportions. Overall, *An. gambiae* accounted for 53.68% of all specified mosquitos, followed by *An. coluzzii* (45.88%) and *An. Arabiensis,* which was found at a very low frequency. This result is similar to the one of Aikpon et al. [33] that showed the same trend in Atacora department, the cotton zone of Benin. The opposite result, displaying *An. coluzzii* as the predominant species, was observed by Gnanguenon et al. [30] in Allada, Dassa, and Malanville. Although these species often have different geographical and seasonal distributions, they can also live in sympatry. In the present study, *An. arabiensis* was found only in Savè, a wet savanna area. This is consistent with works by Gnanguenon et al. [30], who found *An. arabiensis* in Dassa, a commune that is quite close to Savè. Over past years, *An. arabiensis* was considered a dry savanna species found in abundance between the extreme north of Benin and Parakou [34]. It is, therefore, necessary to assess if the species migrated from its preferred area to a more southern area. A possible migration could be explained by factors such as climate change, the destruction of the vector’s ecological zones for agricultural and hunting purposes, the harvesting of timber, and pastoralism.

The findings of the present study revealed spatial heterogeneity in the distribution of *Plasmodium* species prevalent in Benin. The mean prevalence of antibodies of *P. falciparum*, *P. vivax* 210, and *P. vivax* 247 in *An. gambiae* s.l. were, respectively, 2.6%, 0.30%, and 0.2% in the study area. In Cotonou, the three *Plasmodium* species were found in *An. coluzzii* with a high prevalence for *P. vivax 247* (4%) compared to *P. vivax* 210 (2%) and *P. falciparum* (2%). In the communes of Aguégués, Porto-Novo, Ouidah, Agbangnizoun, Athiémé, Karimama, and Sinendé, *P. falciparum* and *P. vivax* 210 were detected with similar frequencies across communes (*p* > 0.05). In Sô-Ava and Dogbo, only *P. falciparum* and *P. vivax 247* were registered with similar frequencies (*p* > 0.05). In the rest of the communes, *P. falciparum* was the sole species found except for Pobè, where no infection to three searched *Plasmodium* species was observed. The null infection observed in Pobè could be due to the low number of mosquitos tested in this commune. This variation of the prevalence of the *Plasmodium* species across communes can be explained by the period of collection of the mosquitoes, as well as the ecological and evolutionary factors associated with mosquitoes, and that can promote divergence and restrict gene flow between vector-adapted parasite lineages [35]. Furthermore, contrary to the findings of Sandeu et al. [36], no co-infection with different species of *Plasmodium* cells was recorded in *Anopheles* mosquitoes analyzed in the communes surveyed over the present study. In addition, over the trial of Sandeu et al. [36] where the different species of *Plasmodium* were searched and quantified in malaria vectors, *P. vivax* was not found in any of the mosquito samples analyzed, contrary to what is shown in the present study. Indeed, Howes et al. [37] previously showed that *P. vivax* was the most prevalent malaria parasite globally but clearly "rare" in Africa. The majority of African populations do not express the Duffy blood group antigen, which is the only known receptor for the *P. vivax* infection. Since this discovery in the 1970s, the low clinical incidence of *P. vivax* in Africa led to a perception that this *Plasmodium* species is completely absent, and that all apparent cases are from a misdiagnosis. Thus, no public health consideration is made for this parasite in terms of diagnosis, treatment, or surveillance. As more sensitive diagnosis methods become available, *P. vivax* infection in Africa is increasingly being reported through various types (entomological, serological, community prevalence) of surveys, as well as clinical data on infections from local residents and international travelers.

It would have been interesting to separate the infection data of mosquitoes according to the biting locations (indoor and outdoor), which we did not do during the trial. Thus, it would be important to take into account the microclimatic conditions (ambient temperature and relative humidity) of the indoor and outdoor environment where collections were undertaken as it matters to the vector availability and density. This constitutes a limitation for the study.

In the present study, the distribution of the different *Plasmodium* species in the sibling species of *An. gambiae* s.l. was also assessed across the surveyed communes. Of the 115 samples positive for *P. falciparum* through PCR, *An. gambiae* was predominant (*n* = 74), followed by *An. coluzzii* (*n* = 40) and *An. arabiensis* (*n* = 1). The infectivity of *An. coluzzii* to *P. vivax* (variant 210 and 247) was significantly higher than that of *An. gambiae*. Indeed, most (76.92%) and all mosquitoes that were positive for *P. vivax* 210 and *P. vivax* 247, respectively, were *An. coluzzii*. This result contradicts findings from Goupeyou-Youmsi et al. [38] obtained in two neighboring villages of the rural commune of Andriba in Madagascar. Indeed, they found that *An. arabiensis,* which was the sole species detected within the *An. gambiae* complex, carried *P. falciparum*, *P. vivax*, and *P. malariae*. Contrary to the present study, only *An. arabiensis* was the vector of three *Plasmodium* species (*P. falciparum*, *P. vivax*, and *P. malariae*). These differences could be explained by weather conditions, the ecological environment, and/or the geographical distribution of vectors that are different between countries [38]. These results suggest that more screening of *An. coluzzii* has to be undertaken, as *P. vivax* is difficult to eliminate, and more ultra-sensitive methods are required to detect such infections in the community. This is a valid point that epidemiologists and entomologists need to worry about since the urban malaria vector, *An. stephensi,* which invaded many parts of sub-Saharan Africa, such as Ethiopia and Sudan [39,40], Somalia, and more recently Nigeria [41], could also be infected with *P. vivax*. For that, the larval habitats of the malaria vectors in Benin need to be documented as efforts for larval source management, along with adult control interventions (LLINs and targeted IRS) that can be strengthened in order to have positive results.

In the present study, PCR allows for the detection of sporozoites of both *P. falciparum* and *P. vivax* in some mosquitoes, while ELISA-CSP gave negative results. A possible explanation would be that the number of sporozoites present in the salivary glands of the mosquito is too low to be detected by the ELISA-CSP test. In addition, several studies have also mentioned the difficulty of ELISA in detecting the antigens of other *Plasmodium* species such as *P. malariae* and *P. ovale*, but also the problem of antigenic variation of the CSP antigen between geographical areas [42,43]. Finally, an extension of this study in all geographical areas of the country and other West African regions could help generate valuable epidemiological data on the two variants of *P. vivax* and allow for a more elaborated malaria pre-elimination policy. The evaluation of *P. vivax* variants is a prerequisite to the development of an effective vaccine. Other aspects that are worth evaluation include the geographical distribution the variants, the intensity of their transmission, the ability of the vectors to allow for their development, and the variability of the human immune response towards them.

## 5. Conclusions

The present study constitutes a first in Benin that detects *P. falciparum* and *P. vivax* in *An. gambiae* s.l. using both CSP ELISA and PCR techniques. The results showed that the risk of exposure of populations to *P. falciparum*, *P. vivax* 210, and *P. vivax* 247 varies from species to species. In addition, both variants of *P. vivax* are present in Benin. It would, therefore, be necessary to evaluate the impact of currently used antimalarial drugs on this *Plasmodium* species for a better therapeutic management of malaria in the country. These results will help the National Malaria Control Program to better plan for the therapeutic management strategy of malaria cases to move towards the pre-elimination of the disease.

## Figures and Tables

**Figure 1 insects-14-00231-f001:**
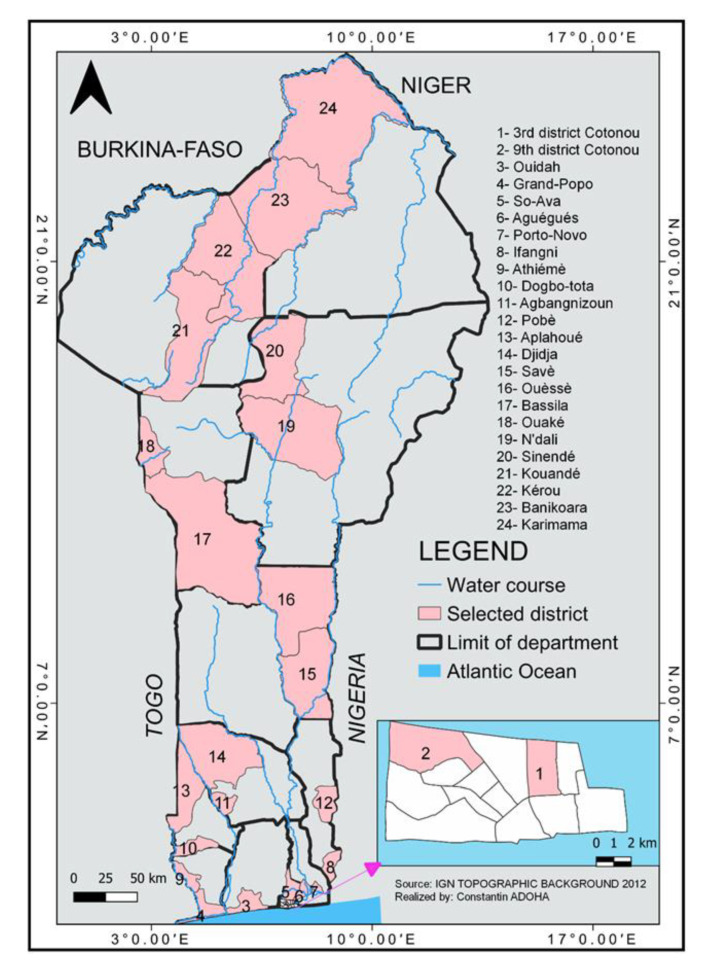
Map of Benin showing the different study sites.

**Figure 2 insects-14-00231-f002:**
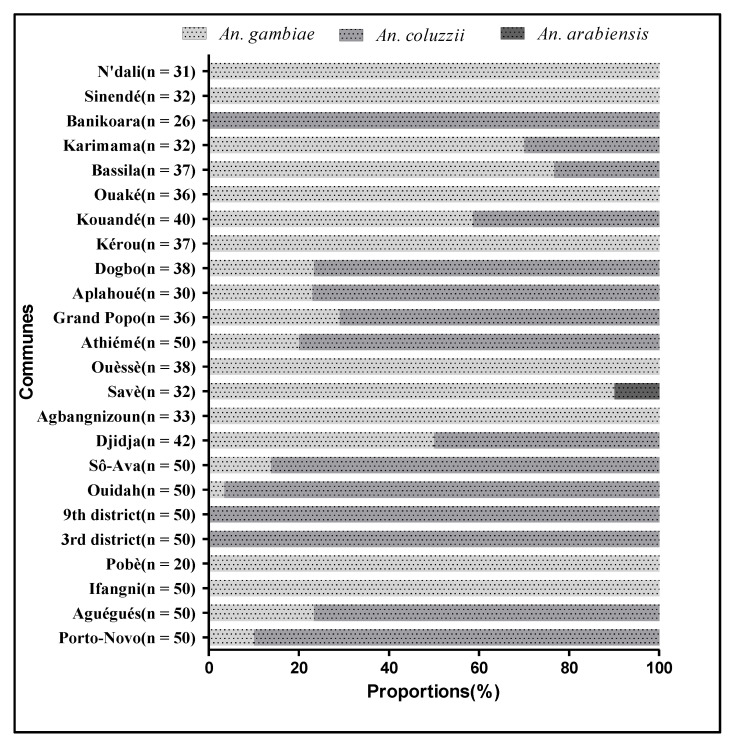
Proportion of the species of the *An. gambiae* complex in each commune.

**Figure 3 insects-14-00231-f003:**
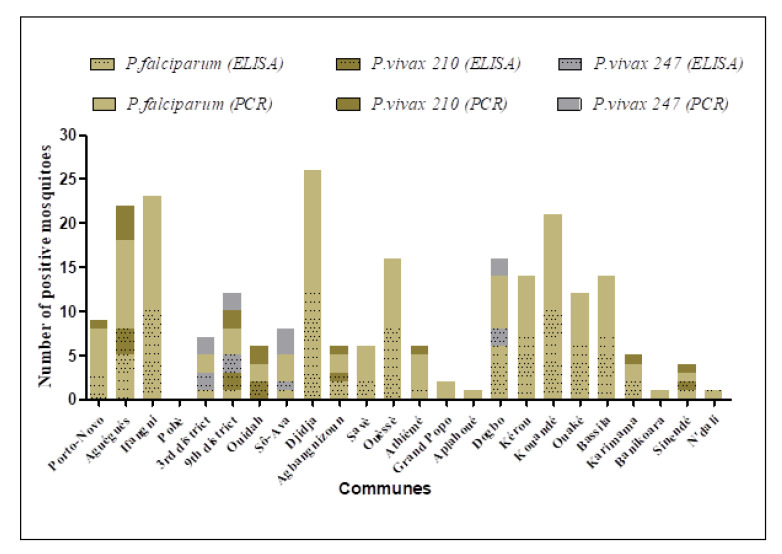
Number of *P. falciparum*, *P. vivax* 210 and *P. vivax* 247 sporozoite positive mosquitoes detected through PCR and CSP ELISA tests in all surveyed communes.

**Figure 4 insects-14-00231-f004:**
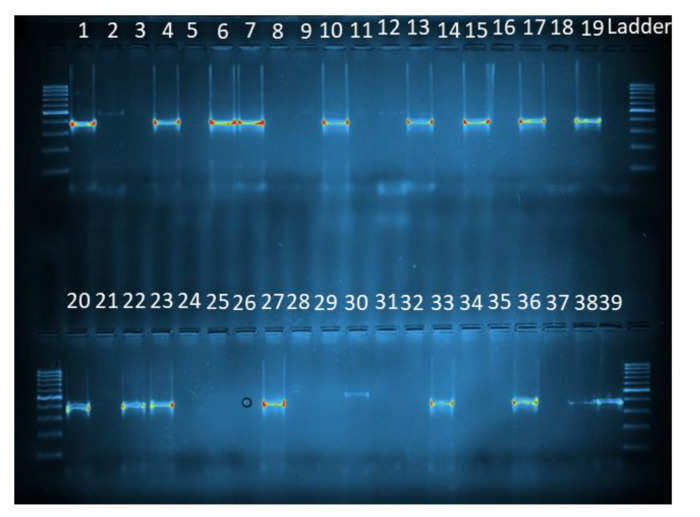
Diagnostic PCR using the primers set for *P. falciparum* and *P. vivax*. 1: positive control *P. falciparum* (395 bp); 2: positive control *P. vivax* (499 bp); 3: negative control; 4–39: samples; Ladder = 100 bp (Solis BioDyne).

**Figure 5 insects-14-00231-f005:**
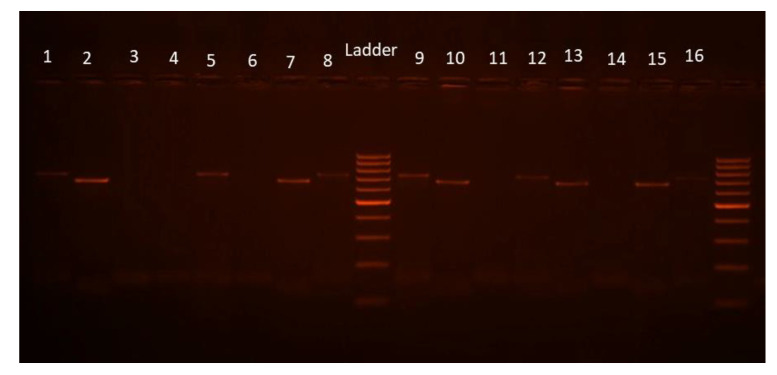
Diagnostic PCR using the primers set for *P. vivax* variants. 1: positive control *P. vivax* 210 (789 bp); 2: positive control *P. vivax* 247 (834 bp); 3: negative control; 4–16: samples; Ladder = 100 bp (Solis BioDyne).

**Table 1 insects-14-00231-t001:** Primers used for the detection and identification of *Plasmodium* and *An. gambiae* complex species.

Species	Primers	Sequences (5′-3′)	Amplicon Size (bp)
Universal reverse *Plasmodium* spp.	UNR PLF	GACGGTATCTGATCGTCTTC AGTGTGTATCAATCGAGTTTC	
*P. falciparum*	FAR	AGTTCCCCTAGAATA GTTACA	395
*P. vivax*	VIR	AGGACTTCCAAGCCGAAGC	499
*P. malariae*	MAR	GCCCTCCAA TTGCCT TCT G	269
*P. ovale*	OVR	GCATAAGGAATGCAAAGAACAG	436
*P. vivax* 210	PVCSP1	CCAGTGCTATGGAAGTTCGTC	789
*P. vivax* 247	PVCSP2	CCAATTTTCCTGTTTCCCATAA	834
*An. coluzzii*	200X6.1F	TCG CCT TAG ACC TTG CGT TA	479
*An. gambiae*	200X6.1R	CGC TTC AAG AAT TCG AGA TAC	249
*An. arabiensis*			223

**Table 2 insects-14-00231-t002:** Culicidae diversity and abundance in the departments of Benin from April to December 2019.

	Ouémé		Plateau		Littoral		Atlantique		Colline		Zou		Mono		Couffo		Atacora		Donga		Alibori		Borgou		Total
Species	Porto-Novo	Aguégués	Ifangni	Pobè	3rd District	9th District	Ouidah	Sô-Ava	Savè	Ouèssè	Djidja	Agbangni Zoun	Athiémé	Grand Popo	Aplahoué	Dogbo	Kérou	Kouandé	Bassila	Ouaké	Karimama	Banikoara	N’dali	Sinendé	
*Anopheles gambiae* s.l.	524	909	490	20	116	577	169	391	99	280	340	93	1469	39	82	169	170	321	174	166	89	19	78	73	6857
*Anopheles funestus* gr	0	0	0	0	0	0	8	0	682	299	17	9	6	0	28	60	34	29	55	40	16	9	0	0	1292
*Anopheles nili* gr	0	0	0	0	0	0	0	0	0	0	0	0	0	0	0	0	187	2	0	0	0	0	0	0	189
*Anopheles pharoensis*	7	16	0	2	0	2	2	83	1	1	0	1	16	3	0	0	3	0	0	0	5	0	0	1	143
*Anopheles ziemanni*	2	2	0	28	0	0	0	12	3	0	0	0	1	10	0	2	30	2	2	15	18	1	0	1	129
*Aedes aegypti*	30	2	3	1	68	70	334	113	116	57	1	14	0	12	11	16	2	0	3	1	1	0	0	0	855
*Other Aedes*	0	0	43	34	0	0	1	6	1	3	1	4	3	0	1	12	6	3	1	2	0	0	0	0	121
*Culex quinquefasciatus*	817	2291	364	806	1736	1419	194	3597	17	5	60	82	622	425	182	65	11	118	35	41	118	58	98	105	13,266
*Other Culex*	24	126	8	22	2	22	2738	550	8	0	15	13	22	309	3	5	1	8	3	22	4	0	12	0	3917
*Mansonia africana*	149	1116	17	2607	1	0	2	457	9	31	1	27	319	599	8	74	22	5	16	20	443	13	3	49	5988
Total	1553	4462	925	3520	1923	2090	3448	5209	936	676	435	243	2458	1398	315	403	466	501	289	309	694	100	191	229	32,757

Other Aedes: *Ae. vitattus*, *Ae. circumluteolus*, *Ae. palpalis gr*, *Ae. Albopictus*; Other *Culex*: *Cx. gr decens*, *Cx. nebulosus*, *Cx. tigripex*, *Cx. annulioris*. *Mansonia uniformis* was collected in low numbers (*n* = 16).

**Table 3 insects-14-00231-t003:** Blood meals source in *An. gambiae* s.l. females collected in Benin departments.

			Blood Meal Source in *An. gambiae* s.l. *females*						
Departments	Communes	Total Tested	N. Human (%)	N. Cow (%)	N. Sheep (%)	N. Pig (%)	N. Human + Cow (%)	N. Human + Sheep (%)	N. Human + Pig
Ouémé	Porto-Novo	5	5 (100)						
	Aguégués	59	58 (98.31)			1 (1.69)			
Plateau	Ifangni	37	37 (100)						
	Pobè	0							
Littoral	3rd district Cotonou	1	1 (100)						
	9th district Cotonou	8	8 (100)						
Atlantique	Ouidah	13	13 (100)						
	Sô-Ava	24	21 (87.5)			2 (8.33)			1 (4.17)
Zou	Djidja	26	26 (100)						
	Agbangnizoun	6	6 (100)						
Collines	Savè	2	2 (100)						
	Ouèssè	1	1 (100)						
Mono	Athiémé	62	58 (93.55)		3 (4.84)	1 (1.61)			
	Grand Popo	2	2 (100)						
Couffo	Aplahoué	5	5 (100)						
	Dogbo	13	13 (100)						
Atacora	Kérou	2	2 (100)						
	Kouandé	6	5 (83.33)	1 (16.67)					
Donga	Ouaké	7	7 (100)						
	Bassila	1	1 (100)						
Alibori	Karimama	28	23 (82.14)	2 (7.14)	1 (3.57)		2 (7.14)		
	Banikoara	0							
Borgou	Sinendé	2	2 (100)						
	N’dali	1	1 (100)						
Total		311	297 (95.5)	3 (0.96)	4 (1.29)	4 (1.29)	2 (0.64)	0	1 (0.32)

N.: Number of *Anopheles gambiae* s.l having taken their blood meal on.

**Table 4 insects-14-00231-t004:** Distribution of *Plasmodium* species in the sibling species of *An. gambiae* complex.

*Anopheles gambiae* s.l. Species	N. Tested	*P. falciparum*	*P. vivax* 210	*P. vivax* 247
*An. gambiae*	463	74	3	0
*An. coluzzii*	474	40	10	9
*An. arabiensis*	3	1	0	0
Total	940	115	13	9

## Data Availability

Data is contained within the article.

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
