# Peer review of "Evidence of Transmission of Plasmodium vivax 210 and Plasmodium vivax 247 by Anopheles gambiae and An. coluzzii, Major Malaria Vectors in Benin/West Africa"

_insects, 2023, doi:10.3390/insects14030231_

Round 1

Reviewer 1 Report

The study is very important because showed more data about case P. vivax infection occurring in region of Africa in important mosquito vectors. The collections were done in many villages of Benin and the number of mosquitoes which were collected was significant.

However, I found a lot problem with reference citations, which needs must be improved and I suggested to included another specific references about the theme.

There are also a lot suggest in the method section, which can improve our manuscript.

And I really think it is important you added information about total collected and total analyzed in each result, such as ELISA and PCR assays. And other point that it is important is data about indoors and outdoors mosquito collected.

In the discussion section I added more suggestions and important points that should be included.

I made comments and added suggestions in the PDF file that is attached to my evaluation.

Author Response

Comments and Suggestions for Authors

The study is very important because showed more data about case P. vivax infection occurring in region of Africa in important mosquito vectors. The collections were done in many villages of Benin and the number of mosquitoes which were collected was significant.

However, I found a lot problem with reference citations, which needs must be improved and I suggested to included another specific references about the theme.

Response: We have done it

There are also a lot suggest in the method section, which can improve our manuscript.

Response: We have done it

And I really think it is important you added information about total collected and total analyzed in each result, such as ELISA and PCR assays. And other point that it is important is data about indoors and outdoors mosquito collected.

In the discussion section I added more suggestions and important points that should be included.

I made comments and added suggestions in the PDF file that is attached to my evaluation.

Response: We have taken into account your comments and suggestions in the current version of the manuscript and have provided some responses in the PDF file.

Many thanks

Reviewer 2 Report

In introduction, sentence reframing is required in line nos. 36 to 39. The statement in line nos. 43 to 45 needs re-evaluation as in addition to P. knowlesi, there are other simian malarial parasites such as P. cynomolgi, P. inui, P. coatneyi in Southeast Asia (Yap et. al., 2021 published in Emerging Infectious Diseases) besides, P. simium and P. brasilianum in South America. Further the authors go on explaining in line nos. 49-52 that P. malariae causes relapse. If so, the reference may be quoted. The reference nos. 13 and 14 are for P. falciparum and P. vivax and not for P. malariae. Sentence reframing is required in line number 53.

In material and methods, Plasmodium referred in line no. 81 need to be italicised. The authors have mentioned about two collection methods, HLC and PSC. Why resting collections were not done in the study sites? It is mentioned that the collectors themselves have collected mosquitoes landing on their legs. If so, there is always a chance of an error in collecting all the mosquitoes as it could be missed if it lands on body parts such as face, making it impossible for the collector cum bait to catch it but to shoo away. Moreover, the collection time documented is from 9 pm to 5 am. It is quite obvious that people will be active during the first quarter of the night and there are vectors which start biting as soon as it becomes dusk, sometimes could be also their peak biting time. Don’t you get vectors biting during this time? Was it a convenience collection method followed? The sweat has many luring constituents and carbon dioxide exhalation lures the mosquito to bite even during the twilight period. How the authors will justify to this pertinent point if they miss to collect during the initial quarter of the night is uncertain. It is also surprising to see the authors winding up collection at 5 am instead of searching till 6 am or sunrise. One collector per night, whether indoor or outdoor from 9 pm to 5 am will also affect the collection efficiency of the collectors at night with a tendency to fall asleep generally after midnight. Rather the authors could have had an additional collector since the mosquito density was also high as documented in Table 1. In figure 1, it is mentioned ‘Watter Course’. Did the authors mean Water course?

Furthermore, the authors also do mention about PSC in bedrooms. What type of houses were selected for PSC? What was their roof characteristics? Roof type of a house do really impact the density of vector mosquitoes. Besides, fed mosquitoes would prefer to rest in an ideal temperature conducive, resting locations with less disturbance. The authors have mentioned in line no. 110 that adult mosquitoes were morphologically identified. It is surprising to note that they didn’t do molecular detection to re-confirm the sibling species of Anopheles gambiae complex? In any field scenario, identification will have to be undertaken with morphological keys followed by confirmation using molecular tools especially in the case of species complexes. This aspect is completely missing in the manuscript. Another important aspect is recording the microclimatic conditions (ambient temperature and relative humidity) of the indoor and outdoor environment where collections were undertaken (HLC or PSC) as it matters the vector availability and density. Why the authors have mentioned beef is still puzzling? The appropriate word for a domestic bovine is not beef.

In the ethical paragraph (lines 169-172), the authors have mentioned that the collectors were regularly followed up? How they were monitored?  More importantly, in Table 1, fairly good number of Aedes aegypti have been collected. What about the precautionary measures to prevent Dengue infection since Ae. aegypti is the primary vector.  It is well known that Ae. aegypti is a dangerous vector with trans-ovarial transmission of viral infections. Moreover, it has not been mentioned about the type of malaria chemoprophylaxis extended to the collectors since they have collected quite a good number of malaria vectors, capable of transmitting malignant form of malaria (Pf). Did any collector become malaria positive while undertaking the study? Did you use the same collectors cum baits for both the seasons as there will be person to person variations in sweat release and carbon dioxide exhalation which aids in mosquito luring resulting in an error if different collector cum baits are used for both dry and wet seasons.

The main focus of the manuscript is about malaria vectors and it’s incrimination rate. Hence, table 1 should be designed in such a way that anopheline species gets a special focus and the other mosquitoes classified as Culicines and/or Aedines. The mosquito species with less numbers could be mentioned in the foot note of the table rather than documenting it in zeros and/or in single digits. Only 940 specimens within the Anopheles gambiae complex have been analysed (line no.191). So what happened to the remining An. gambiae specimens recorded in table 1? Beef mentioned in Table 2 may be replaced with appropriate terms rather than mentioning the flesh of an adult bovine.

In the results section, under the heading PCR test, it has been reported that both PCR and CS-ELISA test was performed. However, it was not mentioned why both the tests were done? If it was to re-confirm then only the method which was sound enough or standardized as explained in the last part of discussion should have been followed. It’s alright to perform both to re-confirm but then a table indicating the discrepancies should have been included. Did the authors find any discrepancy? In the absence of such clarity the authors are left to guess the differences. It is quite obvious from line nos. 257-260 that there is a difference and this information is important to record. Sporozoite positivity has to be correlated with malaria incidence/prevalence. In reference to the statement made in line nos. 250-251, how many specimens were collected from Pobe and out of them what proportion has been screened for vector incrimination? Under the sub-heading of ‘distribution of Plasmodium species in the sibling species of An. gambiae complex’, there is lack of clarity about detection of malarial positivity and its proportion. It looks like P. falciparum was incriminated predominantly in An. gambiae s.l. and P. vivax in An. coluzzii. Does that mean Pv transmission is also going on simultaneously? If so, then more screening of An. coluzzii has to be undertaken as Pv is rather difficult to eliminate and more ultra-sensitive point-of-care detection methods are required to detect such infections in the community.  This is a valid point which epidemiologists and entomologists need to worry since urban malaria vector, An. stephensi has invaded many parts of SSA. Currently, adequate literature is available on the invasion of Anopheles stephensi in the Horn of Africa (2012), Ethiopia and Sudan (2016), Somalia (2019) and Nigeria (2020). It is indecisive if such a situation prevails in Benin or not. So, it is extremely important to screen the mosquito specimens thoroughly involving molecular detection tools as there is always a chance of mis-identification. The breeding habitats of the malaria vectors in Benin need to be documented as efforts for larval source management along with adult control interventions can be strengthened in such areas. The usage of LLIN and IRS coverage in random houses were vectors were collected are also important to find the missing link. Answers to the above queries unfortunately is missing in the manuscript.

Author Response

Reviewer 2: Author’s response to Comments and Suggestions

1- In introduction, sentence reframing is required in line nos. 36 to 39.

Response: We have done it

2- The statement in line nos. 43 to 45 needs re-evaluation as in addition to P. knowlesi, there are other simian malarial parasites such as P. cynomolgiP. inuiP. coatneyi in Southeast Asia (Yap et. al., 2021 published in Emerging Infectious Diseases) besides, P. simium and P. brasilianum in South America.

Response: Thank you for this relevant observation which has been taken into account. Indeed, we have now mentioned the other parasite species responsible for simian malaria. However, we did not add P. simium and P. brasilianum as both species are not yet found in humans. These are species that infect primates and are close to P. vivax and P. malariae.

3- Further the authors go on explaining in line nos. 49-52 that P. malariae causes relapse. If so, the reference may be quoted. The reference nos. 13 and 14 are for P. falciparum and P. vivax and not for P. malariae. Sentence reframing is required in line number 53.

Response: This is corrected

4-In material and methods, Plasmodium referred in line no. 81 need to be italicised.

Response: This is corrected

5-The authors have mentioned about two collection methods, HLC and PSC. Why resting collections were not done in the study sites? It is mentioned that the collectors themselves have collected mosquitoes landing on their legs. If so, there is always a chance of an error in collecting all the mosquitoes as it could be missed if it lands on body parts such as face, making it impossible for the collector cum bait to catch it but to shoo away. Moreover, the collection time documented is from 9 pm to 5 am. It is quite obvious that people will be active during the first quarter of the night and there are vectors which start biting as soon as it becomes dusk, sometimes could be also their peak biting time. Don’t you get vectors biting during this time? Was it a convenience collection method followed? The sweat has many luring constituents and carbon dioxide exhalation lures the mosquito to bite even during the twilight period. How the authors will justify to this pertinent point if they miss to collect during the initial quarter of the night is uncertain. It is also surprising to see the authors winding up collection at 5 am instead of searching till 6 am or sunrise. One collector per night, whether indoor or outdoor from 9 pm to 5 am will also affect the collection efficiency of the collectors at night with a tendency to fall asleep generally after midnight. Rather the authors could have had an additional collector since the mosquito density was also high as documented in Table 1. In figure 1, it is mentioned ‘Watter Course’. Did the authors mean Water course?

Response :

- Actually, collection of resting mosquitoes was performed through PSCs.

- As part of the present study, the way HLCs were performed does not matters much, as the objective of the trial was not to assess the frequency of the mosquito biting rate, and time. However, it has been generally admitted that legs are the part of the body which attract most the mosquitoes. This means that collection of mosquitoes on legs can help having a good estimate of the density, though there could be some minimal errors.

- Regarding the use of an additional collector, actually we did that but, we failed to mention it in the first submitted version. Now, it is done in the revised one (Look, the mosquito collection section, second paragraph).

- Regarding the time at which the collection ended, we recognize that it is a good a suggestion to continue for one extra hour. This is a limitation for the present study.

- Your remark on figure 1 is relevant and now taken into account.

6- Furthermore, the authors also do mention about PSC in bedrooms. What type of houses were selected for PSC? What was their roof characteristics? Roof type of a house do really impact the density of vector mosquitoes. Besides, fed mosquitoes would prefer to rest in an ideal temperature conducive, resting locations with less disturbance.

Response: The houses that were selected have ceilings, and do not have eaves. They also have windows and doors that are kept closed before spraying, which prevents mosquitoes from escaping. PSCs are carried out early in the morning when residents woke up from 7 a.m.

7- The authors have mentioned in line no. 110 that adult mosquitoes were morphologically identified. It is surprising to note that they didn’t do molecular detection to re-confirm the sibling species of Anopheles gambiae complex? In any field scenario, identification will have to be undertaken with morphological keys followed by confirmation using molecular tools especially in the case of species complexes. This aspect is completely missing in the manuscript.

Response : The Morphological (Method section, last paragraph of mosquito collection subsection, lines 132-133) as well as the molecular (Method section, Sub-section of Molecular species identification in the Anopheles gambiae complex, lines 227-238) identification of mosquitoes have been indeed performed as part of the present trial. Moreover, the results of the molecular species ID are mentioned on lines 273-282)

8- Another important aspect is recording the microclimatic conditions (ambient temperature and relative humidity) of the indoor and outdoor environment where collections were undertaken (HLC or PSC) as it matters the vector availability and density. Why the authors have mentioned beef is still puzzling? The appropriate word for a domestic bovine is not beef.

Response : - We agree with you that microclimatic conditions (ambient temperature and relative humidity) as well as the indoor/outdoor environment can have an impact on the mosquito density. Unfortunately we failed to collect these data over the trial. This is a limitation for the study, which is mentioned in the discussion, lines 420-423.

  • For a domestic bovine, the appropriate word is used in the current manuscrit

9- In the ethical paragraph (lines 169-172), the authors have mentioned that the collectors were regularly followed up? How they were monitored?  More importantly, in Table 1, fairly good number of Aedes aegypti have been collected. What about the precautionary measures to prevent Dengue infection since Ae. aegypti is the primary vector.  It is well known that Ae. aegypti is a dangerous vector with trans-ovarial transmission of viral infections. Moreover, it has not been mentioned about the type of malaria chemoprophylaxis extended to the collectors since they have collected quite a good number of malaria vectors, capable of transmitting malignant form of malaria (Pf). Did any collector become malaria positive while undertaking the study? Did you use the same collectors cum baits for both the seasons as there will be person to person variations in sweat release and carbon dioxide exhalation which aids in mosquito luring resulting in an error if different collector cum baits are used for both dry and wet seasons.

Response : They were trained to collect the mosquito before being bitten, and were regularly subjected to medical check-ups. When they had malaria related symptoms, they received anti-malarial medication like Artemisinin-based Combination Therapy. They were also vaccinated against yellow fever.

  • Some collectors had tested positive over the study and were taken care of.
  • We used the same collectors throughout the study period.

  • Regarding dengue fever, we have not taken any particular precautions to prevent this disease in the collectors. By the way, in Benin, the risk of suffering from dengue fever is low even if it exists. Indeed, only one case has been detected and confirmed since 2020 in Benin.

 10- The main focus of the manuscript is about malaria vectors and it’s incrimination rate. Hence, table 1 should be designed in such a way that anopheline species gets a special focus and the other mosquitoes classified as Culicines and/or Aedines. The mosquito species with less numbers could be mentioned in the foot note of the table rather than documenting it in zeros and/or in single digits.

Response : This relevant point was taken into account in the revised version of the manuscript. But we also wanted to show the non-Anopheles mosquitoes that are also vector of other diseases.

11- Only 940 specimens within the Anopheles gambiae complex have been analysed (line no.191). So what happened to the remining An. gambiae specimens recorded in table 1?

Response : Sampling was done in each commune for the analysis of the specimens of Anopheles gambiae s.l.

- For the ELISA CSP testing, a total of 3300 mosquito specimens were analyzed, with on average 150 per commune.

- For the Plasmodium PCR, we sampled 50 Anopheles per commune for the analysis. Indeed, these 50 samples were formed of ELISA positive samples and completed with a sub-sample of negatives ones.

12- Beef mentioned in Table 2 may be replaced with appropriate terms rather than mentioning the flesh of an adult bovine.

Response : This was taken into account

13- In the results section, under the heading PCR test, it has been reported that both PCR and CS-ELISA test was performed. However, it was not mentioned why both the tests were done? If it was to re-confirm then only the method which was sound enough or standardized as explained in the last part of discussion should have been followed. It’s alright to perform both to re-confirm but then a table indicating the discrepancies should have been included. Did the authors find any discrepancy? In the absence of such clarity the authors are left to guess the differences. It is quite obvious from line nos. 257-260 that there is a difference and this information is important to record.

Response : Sometimes, when the amount of parasites is low in the wells, the ELISA test may not detect the positivity of these wells. Thus, PCR which is a DNA detection-based technique increases the likelihood of identifying the parasite species that could not be revealed by the ELISA test. That is the reason for which the PCR technique was used in addition to the ELISA technique.

14- Sporozoite positivity has to be correlated with malaria incidence/prevalence.

Response : This point is relevant. However, as part of the present study, we did not collect incidence/prevalence data in the study communes. That is why, we are not able to do this correlation.

15- In reference to the statement made in line nos. 250-251, how many specimens were collected from Pobe and out of them what proportion has been screened for vector incrimination?

Response : Only 20 specimens were collected in Pobè, and they all were used for the different performed analyzes

16- Under the sub-heading of ‘distribution of Plasmodium species in the sibling species of An. gambiae complex’, there is lack of clarity about detection of malarial positivity and its proportion. It looks like P. falciparum was incriminated predominantly in An. gambiae s.l. and P. vivax in An. coluzzii. Does that mean Pv transmission is also going on simultaneously? If so, then more screening of An. coluzzii has to be undertaken as Pv is rather difficult to eliminate and more ultra-sensitive point-of-care detection methods are required to detect such infections in the community.  This is a valid point which epidemiologists and entomologists need to worry since urban malaria vector, An. stephensi has invaded many parts of SSA. Currently, adequate literature is available on the invasion of Anopheles stephensi in the Horn of Africa (2012), Ethiopia and Sudan (2016), Somalia (2019) and Nigeria (2020). It is indecisive if such a situation prevails in Benin or not. So, it is extremely important to screen the mosquito specimens thoroughly involving molecular detection tools as there is always a chance of mis-identification. The breeding habitats of the malaria vectors in Benin need to be documented as efforts for larval source management along with adult control interventions can be strengthened in such areas. The usage of LLIN and IRS coverage in random houses were vectors were collected are also important to find the missing link. Answers to the above queries unfortunately is missing in the manuscript.

Response : This great contribution is of paramount importance and is included in the discussion of the manuscript. However, some of the points you have raised will be investigated over future studies and will be documented.

Regarding the simultaneous transmission of Pv and Pf by An. coluzzii, we believe this aspect deserves a further scrutiny survey.

We think that we have taken into account the recommendations. We hope that the manuscript will be published soon.

  Many thanks

Reviewer 3 Report

General comment

For any country, to achieve an effective malaria control program, the information on causative parasite diversity and dynamics in the region, is a key factor. This manuscript demonstrates that in Benin the malaria cases occurrence is not solely caused by Plasmodium falciparum (common malaria parasite), and involves other non-falciparum species. Anopheles gambiae s.l (sensu lato) is a major insect-vector responsible for transmitting malaria parasite to human hosts in Benin and therefore the authors established Anopheles gambiae s.l (sensu lato) as a insect-vector of interest to evaluate the prevalence of various Plasmodium species in locally collected mosquito cohorts either through human landing catches and pyrethrum spray catches. This manuscript reports the comparative data on the prevalence of circumsporozoite protein (CSP) antibodies of Pf and P. vivax (Pv) in Anopheles gambiae s.l. in Benin. Author's examine the presence of CSP antibodies in head and thorax of mosquito samples by using ELISA technique as well as evaluate presence of  Plasmodium infection by using universal primers and PCR amplification. Further to delineate the possible  contribution of various Plasmodium species in malaria infection in collected mosquitoes, authors conducted a molecular species specific PCR analysis. The important finding of this study, reveals the variation in the distribution of Plasmodium species prevalences in Benin and the information incurred through this study will be very helpful for malaria control stakeholders, planning and management agency to design effective diagnostics, preventive measures and treatments to curb the lethal impact of malaria in Benin. However, the manuscript needs revision for methods and important results. I have some corrections and comments for authors to consider and that need to be addressed before the manuscript can be acceptable for publication.

Material & methods need some work. This study presents parasite diversity and distribution in countries of interest by using ELISA and PCR diagnostic tests. My major concern,  positive and negative controls for ELISA and PCR tests are missing ? Positive and negative controls serve the purpose of providing evidence that each diagnostic test is successfully performed and is giving the expected level of sensitivity and specificity to compare test samples. Authors need to state controls clearly.

Line 107:  ‘Rambo aerosol spray’ provide company/make 

Line 116; 120-125: Authors must provide which ELISA kit was used to perform the Plasmodium detection. 

I also suggest authors should write the ELISA test procedure briefly for both sections - i) Blood meal origin in An. gambiae s.l collected by PSC ; ii) Circum sporozoite protein ELISA for the detection of P. falciparum, P. vivax 210 and P. vivax 247

It is very useful for readers to find required information, material and methods for future use and to reproduce newer findings. 

Line 117 : Must provide details on what antibodies were used for this study, their make/company, concentrations

Line 142: I suggest changing  ‘DNA amplification’ to’ PCR amplification’

Line 143: Authors can change ‘PCR program’ to ‘PCR conditions’

Line 148-151 : I suggest making a supplementary table for all of the primers used in this study, instead of placing them in text. Also, mention the primers used for section: Molecular species identification in the Anopheles gambiae complex, in this table. 

Line 151-155: I suggest rewriting this section, currently it is unclear how these primers and amplification cycle works.

Line 156-157: Need to rewrite the sentence, authors could state something like - ‘ Total 18 ul of each amplified product for each reaction was analyzed with gel electrophoresis using 2% agarose gel and gel images were recorded for successful amplification. Here, authors needs to state make/company for reagents and instrumentation used.

Line 158-162: Authors need to provide the materials/reagents and PCR conditions used for Molecular species identification in the Anopheles gambiae complex.

In the result section- 

Please combine graphs from Figure 3 and Figure 4, X and Y-axis are the same so it should be easy to combine them and represent with color coding for the number of positive cases detected by ELISA vs PCR respective to various Benin locations. It also gives a better and easy to understand comparative overview of results for readers. 

For PCR amplification, authors need to provide representative gel images ( may be add them to supplementary figures for readers reference). Also authors need to briefly write the amplicon sizes specific to each Plasmodium species

Overall, discussion and conclusions are well written. I suggest authors should make suggested correction to improvise the scientific impact of this intriguing study.

I commend the author's efforts towards conducting such a large-scale sampling, testing with high-end diagnostics and corroborating very important malaria parasite prevalence, diversity and distribution data in Benin. This is a positive step towards effective malaria surveillance and control.

Author Response

Responses to reviewer’s comments

General comment

For any country, to achieve an effective malaria control program, the information on causative parasite diversity and dynamics in the region, is a key factor. This manuscript demonstrates that in Benin the malaria cases occurrence is not solely caused by Plasmodium falciparum (common malaria parasite), and involves other non-falciparum species. Anopheles gambiae s.l (sensu lato) is a major insect-vector responsible for transmitting malaria parasite to human hosts in Benin and therefore the authors established Anopheles gambiae s.l (sensu lato) as a insect-vector of interest to evaluate the prevalence of various Plasmodium species in locally collected mosquito cohorts either through human landing catches and pyrethrum spray catches. This manuscript reports the comparative data on the prevalence of circumsporozoite protein (CSP) antibodies of Pf and P. vivax (Pv) in Anopheles gambiae s.l. in Benin. Author's examine the presence of CSP antibodies in head and thorax of mosquito samples by using ELISA technique as well as evaluate presence of Plasmodium infection by using universal primers and PCR amplification. Further to delineate the possible contribution of various Plasmodium species in malaria infection in collected mosquitoes, authors conducted a molecular species specific PCR analysis. The important finding of this study, reveals the variation in the distribution of Plasmodium species prevalences in Benin and the information incurred through this study will be very helpful for malaria control stakeholders, planning and management agency to design effective diagnostics, preventive measures and treatments to curb the lethal impact of malaria in Benin. However, the manuscript needs revision for methods and important results. I have some corrections and comments for authors to consider and that need to be addressed before the manuscript can be acceptable for publication.

Material & methods need some work. This study presents parasite diversity and distribution in countries of interest by using ELISA and PCR diagnostic tests. My major concern, positive and negative controls for ELISA and PCR tests are missing ? Positive and negative controls serve the purpose of providing evidence that each diagnostic test is successfully performed and is giving the expected level of sensitivity and specificity to compare test samples. Authors need to state controls clearly.

Response: Positive and negative controls were of course used in both analyses (ELISA and PCR). These controls were used to validate the tests performed. The important information requested is completed in the current version of the manuscript on page 4 and 5.

Line 107:  ‘Rambo aerosol spray’ provide company/make 

Response: We have done it

Line 116; 120-125: Authors must provide which ELISA kit was used to perform the Plasmodium detection. 

I also suggest authors should write the ELISA test procedure briefly for both sections - i) Blood meal origin in An. gambiae s.l collected by PSC ; ii) Circum sporozoite protein ELISA for the detection of P. falciparum, P. vivax 210 and P. vivax 247

It is very useful for readers to find required information, material and methods for future use and to reproduce newer findings. 

Response: This important observation has been taken account in the current version of the manuscript

Line 117 : Must provide details on what antibodies were used for this study, their make/company, concentrations

Response: We have done it

Line 142: I suggest changing  ‘DNA amplification’ to’ PCR amplification’

Response: We have corrected it

Line 143: Authors can change ‘PCR program’ to ‘PCR conditions’

Response: We have corrected it

Line 148-151 : I suggest making a supplementary table for all of the primers used in this study, instead of placing them in text. Also, mention the primers used for section: Molecular species identification in the Anopheles gambiae complex, in this table. 

Response: We have done it at page 6

Line 151-155: I suggest rewriting this section, currently it is unclear how these primers and amplification cycle works.

Response: We have corrected it

Line 156-157: Need to rewrite the sentence, authors could state something like - ‘ Total 18 ul of each amplified product for each reaction was analyzed with gel electrophoresis using 2% agarose gel and gel images were recorded for successful amplification. Here, authors needs to state make/company for reagents and instrumentation used.

 Response: We have corrected it

Line 158-162: Authors need to provide the materials/reagents and PCR conditions used for Molecular species identification in the Anopheles gambiae complex.

Response: We have done it

In the result section- 

Please combine graphs from Figure 3 and Figure 4, X and Y-axis are the same so it should be easy to combine them and represent with color coding for the number of positive cases detected by ELISA vs PCR respective to various Benin locations. It also gives a better and easy to understand comparative overview of results for readers. 

Response: We have done it

For PCR amplification, authors need to provide representative gel images (may be add them to supplementary figures for readers reference). Also authors need to briefly write the amplicon sizes specific to each Plasmodium species

Response: We have done it

Overall, discussion and conclusions are well written. I suggest authors should make suggested correction to improvise the scientific impact of this intriguing study.

I commend the author's efforts towards conducting such a large-scale sampling, testing with high-end diagnostics and corroborating very important malaria parasite prevalence, diversity and distribution data in Benin. This is a positive step towards effective malaria surveillance and control.

We think that we have taken into account the recommendations. We hope that the manuscript will be published soon.

  Many thanks

Round 2

Reviewer 1 Report

The authors improved the manuscript and accepted some suggested and explained why didn't accept other. However I still suggestion add information about the the number os mosquitoes collected (lines 259-261; 275; 301). This information is important to know how many mosquitoes can be collected by PSC, and how many mosquitoes the authors analyzed for infections and etc. 

Author Response

The authors improved the manuscript and accepted some suggested and explained why didn't accept other. However I still suggestion add information about the number of mosquitoes collected (lines 259-261; 275; 301). This information is important to know how many mosquitoes can be collected by PSC, and how many mosquitoes the authors analyzed for infections and etc. 

Response : We have included this relevant information. On lines 245 and 248, we mentioned HLC as all the data that have been commented in this sub-section (Mosquito species composition) is solely related to the HLCs. Thus, the number of mosquito collected through HLC is mentioned on line 247.

Of note, the number of mosquitoes collected by PSC was mentioned in lines 276-277.

The other information has been added to lines 263-264, 291 of the current version of the manuscript.

Reviewer 2 Report

I went through the points and found quite a few issues which has been highlighted in red font colour of the attached document. I strongly feel the authors will have to address and incorporate/delete the relevant aspects mentioned to make the manuscript look scientifically good.

Author Response

Responses to reviewer’s comments

I went through the points and found quite a few issues which has been highlighted in red font colour of the attached document. I strongly feel the authors will have to address and incorporate/delete the relevant aspects mentioned to make the manuscript look scientifically good.

Reviewer comments: The last response made by the authors are wrong. Please see the publication of Brasil et.al., published in Lancet Global Health, 2017 wherein the title itself is ‘Outbreak of human malaria caused by Plasmodium simium in the Atlantic Forest in Rio de Janeiro: a molecular epidemiological investigation’. How the authors have missed it is quite strange!

Response: Thank you for your scientific contribution to this paper. This information is now taken into account in the current version of the manuscript in line 60.

Reviewer comments: If HLC was not important, why was it done in the first place? Then the

collections should have been restricted to resting collections (RC) and Pyrethrum spray sheet

collections (PSC). Furthermore, it is true that legs are the most attracting part for biting vectors because legs are exposed unlike other body parts which are covered. What about hands ? This part also attracts mosquitoes. The collectors if they are unable to collect it from hands indicates that either the bait were also collectors, and with frequent hand movement for collection purpose, it is less likely to collect mosquitoes or hands were not targeted for collection. Focusing only on legs means, the collection method adopted was biased and whatever bites on the hands were ignored and therefore less numbers than the actual biting density.

Response: Regarding this part, we didn’t understand each other well. Indeed, we did not mean that the HLC is not important.

Concerning the hands, the density of mosquitoes which bite them is often low. In addition, the collectors we used wore long-sleeved shirts, which considerably reduced the frequency of bites on this part of the collector’s bodies.

Reviewer comments: It is strange to have missed it in the draft and suddenly has been included.

Even with the new numbers, the authors are admitting HLC was not important as mentioned in the

previous paragraph. The punch line is that HLC should be done from 6 pm to 6 am and not from 9 pm

to 5 am whatever may be the reason if you want to include it.

Response: Regarding the collection time, we agree with you that ideally, we should have performed the sampling from 06pm to 06 am. However, we have been obligated to do it from 9pm to 5am due to the budget limitation

Reviewer comments: It is understood that houses have ceilings and do not have eaves. But the

basic question of whether the house has a tiled, thatched, asbestos or concrete roof is unanswered.

Tiled, asbestos and concrete can have ceilings and therefore the type of roof is important as the

temperature fluctuations during day and night in all these roof structures are different, even indoor and outdoors of the same structure.

Response: We agree with you regarding the variation of the temperature in the huts depending on the type of roof. However, it should be pointed out that the collection by PSC was done very early in the morning at 7am. Indeed, at 7:00 am, irrespective of the type of roof, and the season (rainy or dry), the inside of houses is always cool, due to the low morning temperatures. This allows indoor biting mosquitoes to rest inside the houses.

Reviewer comments: Of course, the collectors are trained to collect mosquitoes before being bitten

but ethically chemoprophylaxis has to be administered whether trained or not to avoid chances of getting any malaria symptoms. Administration of malaria drugs need not be waited until the collectors get any symptoms. In an endemic area, the chances of recurring malaria infections being asymptomatic are quite common. What if the collectors were asymptomatic and hence missed to have identified the symptoms? This can lead to occupational based infectious diseases? Therefore, the question of chemoprophylaxis comes into the picture.

Response: In Benin, chemoprophylaxis is not recommended for people living in endemic areas. It is rather recommended for foreigners coming from overseas. So, we complied to the country’s policy. If the collectors are asymptomatic, they will not suffer from the disease. So, they are a reservoir and can develop a certain immunity that allows them not to suffer from the disease. Asymptomatic individuals have no symptom that would identify them as sick.

Reviewer comments: This is ethically wrong if chemoprophylaxis was not administered.

Response: This is not a mistake because this treatment is not allowed for people living in endemic areas according to the policy in force in Benin.

Reviewer comments: This is ethically wrong since quite a few Aedes aegypti was collected

(Table 2) which is the primary vector for Dengue, Chikungunya and Zika.

Response: It is true that specimens of Aedes aegypti were collected in the 12 departments. But due to the very low prevalence of the Aedes borne diseases (Dengue, du Chikungunya et du Zika) in Benin, the likelihood to suffer from them is minimal. However, we agree with you that, special measures should be taken regarding this vector for future studies

Reviewer comments: It is alright if you want to mention other anophelines or mosquito species. But

one has to identify them and not mention as ‘Other Aedes’ and ‘Other Culex’ when the total numbers of such specimens were 121 and 3917 (Table 2). This is fairly good number of mosquitoes unidentified which can be ignored without any identifications. These specimens could have been sent to another institute for identification if the authors could not do it themselves. Any reputed, peer reviewed journal can’t accept it. If the numbers were less, then it is understood unlike this.

Response: I would like point out that all mosquitoes collected in the communes have been morphologically identified using the identification keys. In the first submitted version of the manuscript, we presented a table that showed in detail all mosquito species with their numbers per commune. But you asked us not to use a table with several zeros (n=0) and that only anopheles should be presented. It is following your comment that apart from Anopheles mosquito, we only mention Cx quinquefasciatus, Aedes aegypti, and Mansonia Africana that are very frequent, and grouped all other Culex and Aedes to avoid mentioning several zeros. However, if you are not fine with the table as currently presented, we will bring back the first table submitted to address your concern.

Reviewer comments: Then the authors should have stuck to one method, PCR if the results were

reliable and ignore the other methodology to avoid confusion. If the purpose was to test or compare the efficiency of the detection method, then it is fine or else this was not needed.

Response: Your comment is well noted

Reviewer comments: The prevalence data of the study area is very important to correlate with

vector incrimination results when incrimination of vectors in a particular area is being studied. The data may be collected from the agency/ program and correlate for incorporation in the manuscript. That would make the draft manuscript scientifically better.

Response: There is no data on malaria prevalence in the different surveyed communes in 2019 in Benin. Only incidence data are available. However, the key information on which the present study focusses on, is the different Plasmodium species transmitted by the Anopheles vector, not the clinical data.

We think that we have taken into account the recommendations.

Many thanks

Reviewer 3 Report

Authors have now stated details on Positive and negative controls for both analysis (ELISA and PCR)

Authors have provided company/make for all of the reagents/chemicals/kits used during current study.

Authors have provided ELISA test procedure briefly for both sections - i) Blood meal origin in An. gambiae s.l collected by PSC ; ii) Circum sporozoite protein ELISA for the detection of P. falciparum, P. vivax 210 and P. vivax 247

Authors stated requested details for antibodies used during current investigations

Authors have included all of the primers used for study. However, the table needs formatting, for example if a single primer pair was used to conduct PCR amplification of various mosquito or parasite strains then please merge and place them in the same row, for better readability. 

Authors also worked to improve the results section and made requested additions - Combined graphs from Figure 3 and Figure 4. Authors  included representative gel image for PCR amplification, with the amplicon sizes specific to each Plasmodium species. However the gel image is poorly labeled, and needs proper formatting and labeling for better understanding. Also what about the gel images for  P. falciparum samples ? Need to include a representative gel image.

Authors have made the requested text changes, rewrote the requisite sentences for clarity and corrected all typing mistakes. 

Overall I believe authors have taken in consideration and have made most of the requisite corrections and therefore after making a couple of above mentioned minor corrections I recommend this manuscript for publication.

Author Response

Authors have now stated details on Positive and negative controls for both analysis (ELISA and PCR)

Authors have provided company/make for all of the reagents/chemicals/kits used during current study.

Authors have provided ELISA test procedure briefly for both sections - i) Blood meal origin in An. gambiae s.l collected by PSC ; ii) Circum sporozoite protein ELISA for the detection of P. falciparum, P. vivax 210 and P. vivax 247

Authors stated requested details for antibodies used during current investigations

Authors have included all of the primers used for study. However, the table needs formatting, for example if a single primer pair was used to conduct PCR amplification of various mosquito or parasite strains then please merge and place them in the same row, for better readability. 

Response : We have done it in table 1, line 217.

Authors also worked to improve the results section and made requested additions - Combined graphs from Figure 3 and Figure 4. Authors  included representative gel image for PCR amplification, with the amplicon sizes specific to each Plasmodium species. However the gel image is poorly labeled, and needs proper formatting and labeling for better understanding. Also what about the gel images for  P. falciparum samples ? Need to include a representative gel image.

Response: The gel image has been formatted with appropriate labelling for better understanding in line 345-349.

We have also added a representative gel image for P. falciparum samples in lines 340-344.

Authors have made the requested text changes, rewrote the requisite sentences for clarity and corrected all typing mistakes. 

Overall I believe authors have taken in consideration and have made most of the requisite corrections and therefore after making a couple of above mentioned minor corrections I recommend this manuscript for publication.

Thanks
